# DYAD: A Descriptive Yet Abjuring Density efficient approximation to linear neural network layers

**Sarin Chandy**$^*$   **Varun Gangal** $^*$   **Yi Yang**   **Gabriel Maggiotti**
ASAPP Inc.
{schandy,vgangal,yyang,gmaggiotti}@asapp.com

## Abstract

We devise, implement and performance-asses DYAD, a layer which can serve as a faster and more memory-efficient approximate replacement for linear layers, (*nn.Linear()* in Pytorch). These layers appear in common subcomponents, such as in the *ff* module of Transformers. DYAD is based on a bespoke near-sparse matrix structure which approximates the dense "weight" matrix $W$ that matrix-multiplies the input in the typical realization of such a layer, a.k.a DENSE. Our alternative near-sparse matrix structure is decomposable to a sum of 2 matrices permutable to a block-sparse counterpart. These can be represented as 3D tensors, which in unison allow a faster execution of matrix multiplication with the mini-batched input matrix $X$ compared to DENSE $(O(rows(W) \times cols(W)) \rightarrow O(\frac{rows(W) \times cols(W)}{\# \ of \ blocks}))$. As the crux of our experiments, we pretrain both DYAD and DENSE variants of 2 sizes of the OPT arch and 1 size of the Pythia arch, including at different token scales of the babyLM benchmark. We find DYAD to be competitive ($\geq 90\%$) of DENSE performance on zero-shot (e.g. BLIMP), few-shot (OPENLM) and finetuning (GLUE) benchmarks, while being $\geq$7-15% faster to train on-GPU even at 125m scale, besides surfacing larger speedups at increasing scale and model width.

## 1  Introduction

Riding on the back of the already pivotal decade-long rise of GPU-driven deep learning [1], Transformers [2] in 2017 crescendoed the ambition, scale and task-generality of ML models. With cross-sequence in-training parallelizability and representation power through all-pair interactions, transformers disrupted NLP and its incumbent recurrent paradigm [3], but since became key components in other modalities such as CV [4]. Pretrained models as base representations, limited then to CV, emerged via LLMs like BERT [5], T5 [6] etc reaching SOTA across tasks with limited finetuning.

A natural consequence of a module's ubiquity is that even a small improvement to one of its aspect can have major impact on its application and research — as seen by the recent impact of e.g., quantization [7]. A result of this is that an inefficient component (attention) sees a barrage of research (e.g.hashing [8], softmax alternatives [9], FlashAttention [10] etc) until some other component emerges as a bottleneck. We believe this is the case with the dense linear layers in the Transformer's $ff$ module. Moreover, models have larger hidden dimension (4096 for Pythia, 8192 for Llama2), leading to quadratic rise in compute from *ff* module linear layers. Thus inspired, we devise DYAD (*Descriptive Yet Abjuring Density*) — an efficient linear layer approximation using block-sparsity.

---

$^*$Equal Contribution. Sarin proposed the dyad model, modified the models in the transformers library with the Dyad layer and wrote the formulation section. Varun proposed the evaluation frameworks, organized the experiments and led the paper writing

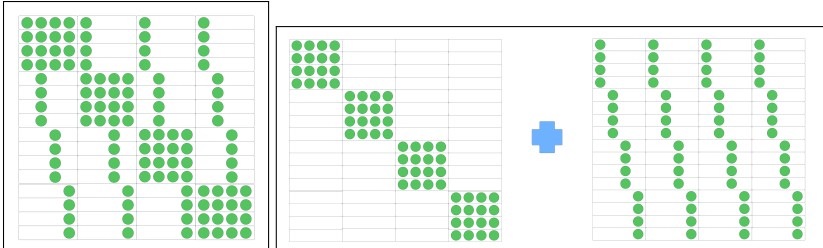

Figure 1: Dyad Weight Matrix [L] vs its Components [R], BLOCKDIAG & BLOCKTRANS. Green is $\neq 0$.

## 2 Formulation

### 2.1 Linear Layer

A Linear layer is the basic building block of all neural networks, represented in pytorch by `nn.Linear()`. It maps input $X$ to output $Y$ via a dense matrix multiplication with weight matrix $W$, given by the equation $Y = G_{Linear}(X) = WX + b$. Here, $W$ is a matrix of shape $f_{out} \times f_{in}$ where $f_{out}$ and $f_{in}$ represent the no. of output & input features. $Y$, $X$ and the bias $b$ have shapes $f_{out} \times n_{batch}$, $f_{in} \times n_{batch}$ and $f_{out} \times 1$. Frameworks like pytorch pose the shape of $X$ and $Y$ as $n_{batch} \times f_{in}$ and $n_{batch} \times f_{out}$ but here we adhere to the former convention.

### 2.2 DYAD : Definition and Properties

We introduce a family of sparse layers named DYAD that can serve as an approximate replacement for the dense linear layer. DYAD has 3 variants called DYAD-IT, DYAD-OT and DYAD-DT. The initials stand for Input Transpose, Output Transpose and Double Transpose. They are named such because transpose operations on either the input or output enables to compute their outputs efficiently. We describe DYAD-IT here and will describe the other two in a later section. DYAD is a linear layer with a sparse weight matrix having shape shown in Fig 1. The output of this layer can be calculated using $G_{Linear}$. However, this won't lead to any efficiency gain compared to the linear layer. We can split the DYAD matrix into 2 components as shown in Fig 1. These components share some non-zero elements but their sum's representational power would be identical to the DYAD matrix. We call the first component the *Block Diagonal Component* (BLOCKDIAG) and the second one the *Block Transposed Component* (BLOCKTRANS). The ability to split DYAD into 2 components is what inspires its name. A DYAD matrix can be defined using 3 parameters, $n_{dyad}$, $n_{in}$ and $n_{out}$. $n_{out} \times n_{in}$ is the size of each submatrix in BLOCKDIAG and $n_{dyad}$ represents the no. of submatrixes in each component. Thus, all the figures for DYAD shown here have $n_{dyad} = n_{in} = n_{out} = 4$. With the 2 components of DYAD split up, we can write its layer output as in Eq 1.

$$Y = W_1 X + W_2 X + b \tag{1}$$

Naively implementing this as in Eq 1, will be as expensive as its dense counterpart. To exploit the joint properties of sparsity and block structure in these 2 components, we need to transform $W_1 X$ and $W_2 X$ to an equivalent sequence of 3D tensor operands and operations.

Hereforth, we ease representing 3D tensors in our equations by overloading pytorch tensor operators.

#### 2.2.1 Efficient Computation of BLOCKDIAG

Let $Y_1 = W_1 X$ be the output of BLOCKDIAG. From Fig 1, we can see that for any $Y_1[i \times n_{out} : (i+1) \times n_{out}, :]$ only depends on $X[i \times n_{in} : (i+1) \times n_{in}, :]$ where $i \in [0, n_{dyad})$. This shows that each pair of $Y_1[i \times n_{out} : (i+1) \times n_{out}, :], X_1[i \times n_{in} : (i+1) \times n_{in}, :]$ can be calculated individually using a matrix multiplication. The weights needed for this are $W_1[i \times n_{out} : (i+1) \times n_{out}, i \times n_{in} : (i+1) \times n_{in}]$. We can store the weights needed for all these pairs of outputs and inputs as a 3D tensor, $W_1'$ of shape $(n_{dyad}, n_{out}, n_{in})$ as per Eq 2.

$$W_1'[i, j, k] = W_1[i * n_{out} + j, i * n_{in} + k] \tag{2}$$

This is a factor of $n_{dyad}$ times smaller when compared to $W_1$ since it has the shape $(n_{dyad} \times n_{out}, n_{dyad} \times n_{in})$. Thus, the whole output of the layer can be computed together with a single batched matrix multiplication as shown in Eq 4 after the input has been also converted to a 3D tensor as

shown in Eq 3.

$$X_1^{'} = X.reshape(n_{dyad}, n_{in}, n_{batch}) \tag{3}$$

$$Y_1 = W_1^{'}.bmm(X_1^{'}).reshape(n_{dyad} \times n_{out}, n_{batch}) \tag{4}$$

The value of $Y_1$ here is the same as $W_1 X$ but cost of computing it will be $O(n_{dyad} \times n_{out} \times n_{in})$ instead of $O(n_{dyad}^2 \times n_{out} \times n_{in})$ which is $O(n_{dyad})$ times faster.

### 2.2.2 Efficient Computation of BLOCKTRANS

The matrix multiplication for BLOCKTRANS, i.e. $W_2 X$ can be converted to a form similar to BLOCKDIAG by permuting the columns of $W_2$. A permutation matrix, $P$ is a square matrix which has exactly one element along each row and each column as one and the rest have a value of zero. Pre-multiplying by a permutation matrix ($PA$), permutes the rows of matrix $A$, while post-multiplying ($AP$), permutes the columns of matrix $A$. So if we post multiply, $W_2$ by an appropriate permutation matrix which has the form as shown in Eq 5, we will end up with a matrix similar to BLOCKDIAG.

$$P(i, j) = \delta_{j = n_{dyad} * (i \% n_{in}) + i // n_{in}} \tag{5}$$

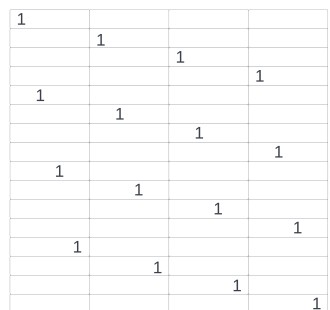

Figure 2: Permutation Matrix for BLOCKTRANS

For Dyad with $n_{dyad} = n_{in} = n_{out} = 4$, this permutation matrix is shown in Fig 2. As Permutation matrices are orthonormal, $P^{-1} = P^T$ where $P^T$ is another Permutation matrix [11].

$$Y_2 = W_2 X = W_2(PP^T)X = (W_2 P)(P^T X) \tag{6}$$

Using the othonormal property of permutation matrixes we can write $W_2 X$ as shown in Eq 6. Here, $Y_2$ is the output of BLOCK-TRANS. Let $W_2^P = W_2 P$ and $X_2^P = P^T X$. Since, $W_2^P$ has the same structure as the weight matrix for BLOCKDIAG, we can also store it as a 3D tensor, $W_2^{'}$, of shape $n_{dyad} \times n_{out} \times n_{in}$. Hence, as in the case before this leads to a reduction in memory size of $O(n_{dyad})$ when compared to $W_2$.

Calculating $P^T X$ naively in order to get $X_2^P$ will be as expensive as the linear layer. However, the specific pattern of the permutation allows us to calculate $X_2^P$ by simplying transposing a 3D view of $X$. We can see from Eq 5 that within a multiple of $n_{in}$ for every increment of $i$, $j$ increases by $n_{dyad}$, while for every $i + n_{in}$ increment j only increases by 1. Thus, $i$ can be thought of as the 1D index of a flattened 2D matrix of shape $n_{dyad} \times n_{in}$ with stride $(1, n_{dyad})$. Hence, permuting and inverting the permutation can be done by just transposing this 2D matrix i.e. going from shape $n_{dyad} \times n_{in}$ to $n_{in} \times n_{dyad}$ and from stride $(1, n_{dyad})$ to stride $(n_{dyad}, 1)$ and vice versa. Calculating $X_2^P$ this way from $X$ is shown in Eq 7.

$$X_2^P = X.reshape(n_{in}, n_{dyad}, -1).transpose(0, 1).reshape(-1, n_{batch}) \tag{7}$$

$$X_2^{'} = X_2^P.reshape(n_{dyad}, n_{in}, n_{batch}) \tag{8}$$

$$X_2^{'} = X.reshape(n_{in}, n_{dyad}, n_{batch}).transpose(0, 1) \tag{9}$$

Now as in the case of BLOCKDIAG we need the activations as a 3D tensor to do the calculations efficiently. So we need to reshape $X_2^P$ as shown in Eq 8 to get $X_2^{'}$. $X_2^{'}$ is the actual activation input for the batched matrix multiplication. We can combine Eq 7 and 8 to cancel the reshape as shown in 9. Eq 9 is basically free and involves just changing some metadata related to the strides of the dimensions, as shown in Fiture 3. The actual data of the tensor need not be touched here.

$$Y_2 = W_2^{'}.bmm(X_2^{'}).reshape(n_{dyad} \times n_{out}, n_{batch}) \tag{10}$$

Finally, the output of BLOCKTRANS can be computed as shown in Eq 10. The cost of computation here is thus, the same as that of the previous case with complexity $O(n_{dyad} \times n_{out} \times n_{in})$ and is faster by a factor of $O(n_{dyad})$ when compared with multiplying with $W_2$. In Appendix §5.3.3, we discuss some thoughts about the representational power of DYAD.

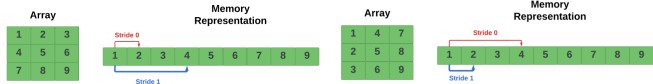

**Figure 3:** Illustrations of BLOCKTRANS computation, in particular the Equation 9 step

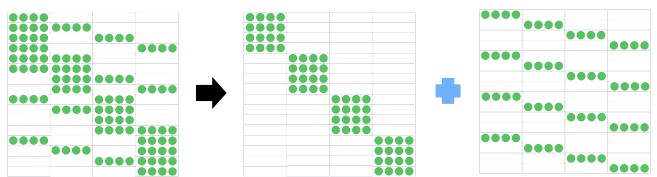

Figure 4: Dyad Output Transposed

## 2.3 DYAD implementation in pytorch

The DYAD layer can be written relatively efficiently with a few lines of code in native pytorch. Here we present a simple implementation of DYAD, more specifically the exemplary DYAD-IT. Note that, in the code, we use $dim$ instead of $n$ to denote dimension. We also note that this code has some overhead in terms of multiple kernel launches, sequential processing of the components and some copying that could be avoided. Even with all of this we are still able to observe significant speedups especially at higher model scales.

```python
class Dyad(torch.nn.Module):
    def __init__(self,shape,bias=True):
        super().__init__()
        self.dyad_dim, self.dim_in, self.dim_out = shape
        self.has_bias = bias
        k = 1.0/float(np.sqrt(dim_in*dyad_dim))
        self.wu = torch.nn.Parameter(torch.empty((dyad_dim,dim_out,dim_in)))
        torch.nn.init.uniform_(self.wu,-k,k)
        self.wl = torch.nn.Parameter(torch.empty((dyad_dim,dim_out,dim_in)))
        torch.nn.init.uniform_(self.wl,-k,k)
        if self.has_bias:
            self.bias = torch.nn.Parameter(torch.empty((dyad_dim*dim_out,1)))
            torch.nn.init.uniform_(self.bias,-k,k)

    def forward(self,x):
        # The shape of x is (dyad_dim x dim_in, batch_size)
        x1 = x.reshape(self.dyad_dim,self.dim_in,-1)
        # The shape of x1, which is a view of x, is now (dyad_dim, dim_in,
            batch_size)
        x2 = x.reshape(self.dim_in,self.dyad_dim,-1).transpose(0,1)
        out =
            (self.wl.bmm(x1)+self.wu.bmm(x2)).reshape(self.dyad_dim*self.dim_out,-1)
        if self.has_bias:
            out+= self.bias
        return out
```

## 2.4 Dyad Variants

In this section we will describe the other two variants of Dyad, Dyad-OT and Dyad-DT. Both of these variants can be split into two components. As in the case of Dyad-IT, the first component is a block diagonal matrix and the second component can be converted back into a block diagonal by means of transposes.

### 2.4.1 Dyad-OT

The weight matrix and the two split components of Dyad-OT is shown in Fig 4. The first component can be calculated exactly the same way as in Dyad-IT. The output of the second component can be calculated as $Y_2 = W_2 X$. Here, $Y_2$ is the output of the second component, $W_2$ is the weight matrix

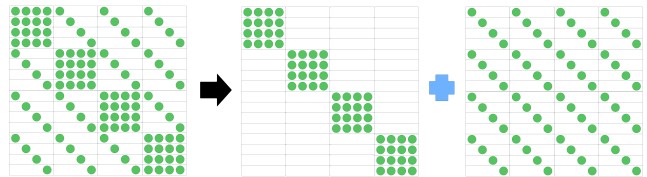

Figure 5: Dyad Double Transposed

and $X$ is the activation. Similar to the case of Dyad-IT, we can see that if we permute the second component along the rows we can get back a block diagonal matrix. Let the permutation matrix which achieves this be $P$. Since, we are permuting the rows here this permutation matrix needs to be pre multiplied i.e $W_2^P = PW_2$ where $W_2^P$ is the resultant block diagonal matrix. We can convert $W_2X$ to use this form as shown below.

$$Y_2 = (P^T P)W_2 X \tag{11}$$

$$Y_2 = P^T (PW_2)X \tag{12}$$

$$Y_2 = P^T W_2^P X \tag{13}$$

Here, we can calculate $W_2^P X$ similar to the first component and then the permutation by permultiplying $P^T$ can be achieved by transposing the output similar to how it was done for Dyad-IT. Thus, similar to Dyad-IT we will have a compute complexity of $O(n_{dyad} \times n_{out} \times n_{in})$.

### 2.4.2 Dyad-DT

Fig 5 shows the weight matrix and the components of Dyad-DT. The important thing to note is that the second component can be converted into a block diagonal matrix through a combination of transposing the cloumns as well as transposing the rows. So, in other words it's basically a combination of Dyad-IT and Dyad-OT. We have to transpose the input before we multiply by the block diagonal weight matrix and then we have to transpose the output to get the final output of the layer.

$$Y_2 = (P_2^T P_2)W_2(P_1 P_1^T)X \tag{14}$$

$$Y_2 = P_2^T (P_2 W_2 P_1)(P_1^T X) \tag{15}$$

$$Y_2 = P_2^T W_2^P X^{'} \tag{16}$$

The above equations show this. $X^{'} = P_1^T X$ is the result of transposing the input while $W_2^P$ is the equivalent block diagonal matrix obtained by permuting both the columns and rows ($P_2 W_2 P_1$). As in the case with the other two variants, this variant also achieves a complexity of $O(n_{dyad} \times n_{out} \times n_{in})$.

As further food for thought, we present a sketch discussing some thoughts about the representational power of DYAD in Appendix §5.4.

## 3 Experimental Setup: Architectures, Benchmarks and Metrics

### 3.1 Choice of Pretraining Corpus

Since our experiments need multiple pretraining runs to create different pretrained variants of the same architectures, each with the linear layers of the *ff* module replaced by our DYAD variants, in addition to the baseline DENSE, it would be infeasible to pretrain manyfold on full corpora, especially for a new method that can show on-the-fly challenges. Since TinyStories [12], there has been an emerging class of lean pretraining corpora (others being [13], [14]) carefully curated to forsake on superficial aspects of scale (e.g. internet-scale vocab), while being linguistically rich enough. They present a reasonable Goldilocks choice, being small enough to pretrain many runs on, while being large enough to learn emergent LLMesque skills. Hence, we choose BABYLM [14], which comes in two scales - 100M and 10M tokens respectively. The authors also provide an easy-to-use and "hackable" setup, with repos that support a) pretraining b) evaluating on BLIMP/GLUE.

### 3.2 Models, Architecture, Hyperparameters & Compute

We seek a setting which allows direct comparison between DENSE vs DYAD, with preferably simple loss function and minimally randomized training. We avoid encoder-only and encoder-decoder

architectures for this reason. To compare with BABYLM baselines, we pick the sole decoder-only architecture they evaluate, i.e. OPT-125m [15], as the architecture to try our variants with. We lay greater emphasis on exhaustive experiments at 10M data scale, though we also perform a core subset at the 100M scale. To show generalization to higher architecture size, we also repeat some experiments with OPT 350-m. We also present promising results at 10M with Pythia 160-M in Appendix §5.6.4. We refer to the pretrained DENSE checkpoint shared from BABYLM as DENSE-EXT, DENSE being our replication of it keeping pretraining details same for DYAD. DYAD variants have $n_{dyad} = 4$ unless mentioned ($-8$ i.e. $n_{dyad} = 8$). All experiments are on 1 GPU. More compute details are noted in Appendix§5.5

### 3.3 Benchmarks & Metrics

**Zero-Shot: BLIMP** Benchmark of Linguistic Minimal Pairs (BLIMP) [16] consists of pairs of grammatical-ungrammatical sentences grouped by 12 broad phenomena e.g. anaphora and noun-verb agreement. A good LLM ought to assign higher probability to the grammatical member.

**Few-Shot: OPENLLM** The OPENLLM leaderboard [17] has become a prevalent way to benchmark LLMs based on 4 few-shot openbook MCQesque benchmarks. Internally, it uses LMEvalHarness [18], which we replicate to compute numbers for our models as well as BabyLm's pretrained checkpoints.

**Finetuned:GLUE+** General Lang. Understanding Eval (GLUE) [19], is a set of 7 NLU tasks, each evaluated post-finetuning. Also, we compute results on WSC and BOOLQ. We christen this GLUE+.

**Training Time** We report both total and *FF*-only (time spent just on *ff* modules) time per minibatch.

**Memory & Parameter Footprint** By storing the dense subset of $W$ as 3D tensor form, DYAD has lesser space complexity. To gauge real space saved, we measure various notions of memory and parameter size:
i) **Non-Embedding Parameters:** As in Pythia [20], we report total Non-Embedding Parameters.
ii) **Model Checkpoint Size:** On-disk size of the model checkpoint.
iii) **In-Training GPU Memory Usage:** During training, models may use memory well beyond parameters, e.g. optimizer state, cached activations etc. In-Training GPU Memory Usage as a metric incorporates this.

### 3.4 Results

#### 3.4.1 DYAD vs DENSE with 10M tokens

Through Tables 2 and 3 (and Appendix Tables 6, 7 & 8), we see that DYAD variants are well competitive ($\leq 5\%$) of the best DENSE baseline.

In addition, through Figures 7, and Tables **??**, 4 and 5 (as well as Appendix Tables 9 and 10), we see that all DYAD variants can translate the better complexity to actual speedups. We see that the quantum of these speedups to be much higher for larger architecture sizes i.e. OPT-350m

#### 3.4.2 Promising Results With Pythia

The Pythia suite [20] of models by EleutherAI, trained based on a permissively licensed collected dataset named The Pile [21].

The results we get by pretraining Pythia on the 10M scale of BABYLM are shared in Table 3. We also see that, just as we did for OPT 125-m, the promised time complexity improvements translate into speedups considering both FF-only time (as we can see in Table 5) and overall time (as we can see in Table 4)

#### 3.4.3 *-CAT experiments

As can be seen in the `forward()` function of our implementation of DYAD-IT laid out in §2.3, and as we note explicitly therewith ("*We also note that this code . . . has some overhead . . . sequential processing of the components*"), having to process the two components underlying our layer, i.e. BLOCKDIAG and BLOCKTRANS in separate steps does introduce an unnecessary overhead that did not exist for DENSE. To mitigate this, one can conceptualize a slightly faster forward layer that first concatenates the two components to enable parallel processing, before adding them up. We refer to implementations which use this variation by the -CAT suffix, such as in DYAD-IT-CAT. We perform a pretraining run of this variant at 10M scale, and find that this is indeed faster as anticipated, while retaining near-identical performance. Specifically, the $ff$-only time per minibatch taken by

DYAD-IT-CAT along with OPT-125m is 3.27 ms, rather than 3.90ms taken by the simple, no -CAT, DYAD-IT, which is about 16% faster. For OPT-350m, the fractional speedup goes up even more, with DYAD-IT-CAT taking 5.46 ms and with DYAD-IT taking 7.92ms, being 45% faster.

The gains seen by optimizing away even this small overhead point to the promise held by the opportunity to optimize other steps of this layer once matrix multiplication itself has been optimized [through using DYAD style layers].

### 3.4.4 Profiling Experiments At Wider Architectural Scales

Since DYAD is primarily applied herein to $ff$ module, assessing its benefits at higher relative width would give us important additional insight on its salience and generalizability in terms of benefit.

To do this, we take the OPT-1.3B model's architecture but cap its depth down to 6 layers so that the model continue to fit within our computational constraints at levels of width all the way upto 4096.

### 3.4.5 Testing Waters with Vision Applicability - MNIST Experiment(s)

Since the bulk of our experiments as well as intuitions and writing is in a large language model/NLP context, a natural question that may perplex a reader is if using a DYAD style linear layer rather than a DENSE one holds promise in other modalities e.g. computer vision. To make a basic probe in this direction, we do experiments with the simple but foundational MNIST digit classification task [22], replacing linear layers with both their plain DENSE and our DYAD-IT, again with $n_{dyad} = 4$. Furthermore, we also test the waters in terms of trying our approaches with diverse accelerators by performing these experiments directly on a Macbook CPU without using a GPU or MPS etc.

We find the properties of reasonable performance preservation and speedups in both ff and overall time carry over to this situation too. Specifically, we find DYAD-IT achieves 98.51% test accuracy vs the 98.43% achieved by DENSE, while taking 3.76 seconds of $ff$-only time per minibatch compared to 4.85 seconds by DENSE.

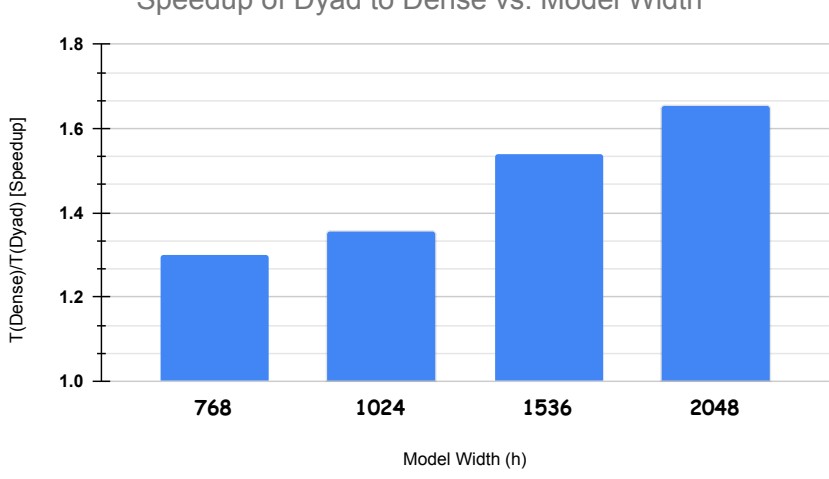

Figure 6: DYAD vs DENSE Speedup At Different Model Widths of 6-Layer Capped OPT-like architecture.

| Model | Forward Pass | Backward Pass | Total | Total speedup ratio |
|---|---|---|---|---|
| DENSE | 1.458818136 | 2.843522568 | 4.302340703 | 1 |
| DYAD-IT | 1.037282137 | 2.864683089 | 3.901965226 | 1.102608674 |
| DYAD-OT | 1.005873492 | 2.833987413 | 3.839860905 | 1.12044181 |
| DYAD-DT | 1.048527787 | 2.955974824 | 4.004502611 | 1.074375802 |
| DYAD-IT-8 | 0.7726907735 | 1.836098994 | 2.608789767 | 1.649171105 |

Table 1: Mean time taken per minibatch by the ff transformer modules of OPT-125m training on account of forward, backward passes and in total. All times are in milliseconds. Speedup ratio is computed w.r.t. DENSE

miu

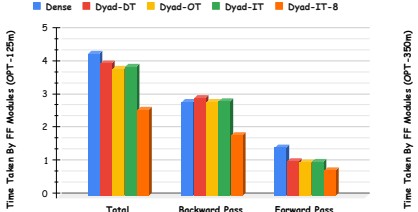 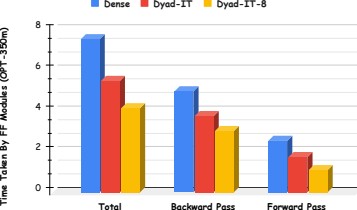

Figure 7: Mean traintime per minibatch by FF modules of OPT-125m/OPT-350m training spent on forward, backward passes and total (Times in ms). DYAD variants are faster, and $\uparrow n_{dyad}$ (DYAD-IT-8) improves this.

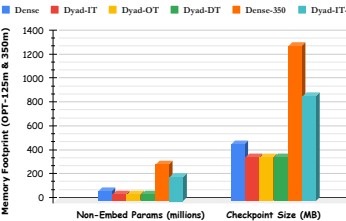 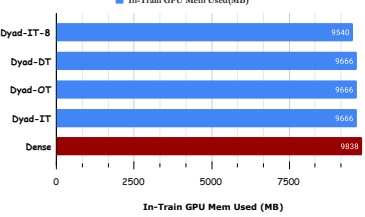

Figure 8: Memory and parameter footprint of OPT-125m/OPT-350m training as per various static estimates on the left and dynamic GPU mem usage on the right.

| Benchmark | Task | DENSE | DENSE-EXT | DYAD-IT | DYAD-OT | DYAD-DT | DYAD-IT-8 |
|---|---|---|---|---|---|---|---|
| GLUE+ Mean | GLUE+ | 68.82 | 63.38 | 67.33 | 68.46 | 68.59 | 67.70 |
| | GLUE+-QA | 66.37 | 63.67 | 66.27 | 66.27 | 63.69 | 64.02 |
| | GLUE+-NLI | 68.27 | 59.78 | 65.64 | 68.27 | 68.67 | 67.65 |
| BLIMP Mean | BLIMP | 59.16 | 60.31 | 60.47 | 62.55 | 60.86 | 58.88 |
| OPENLLM Means | OPENLLM | 30.27 | 30.39 | 30.61 | 30.74 | 30.58 | 30.65 |

Table 2: Performance on GLUE+ (finetuning), BLIMP (0-shot), OPENLLM (few-shot) benchmarks for DENSE baselines vs 3 DYAD variants with $n_{dyad} = 4$ and a sparser version of the 1st (DYAD-IT-8). Numbers which exceed DENSE/DENSE-EXT are bolded/underlined respectively. All DYAD variants are $\geq 0.95 \times max(\text{DENSE}, \text{DENSE-EXT})$. We present aggregates for brevity and defer individual values to Appendix Table 2

| Benchmark | Task | DENSE | DYAD-IT |
|---|---|---|---|
| Means | GLUE+ | 73.86818182 | **73.71942857** |
| | GLUE+-QA | 69.875 | **72.7185** |
| | GLUE+-NLI | 73.188 | **77.2675** |
| Means | BLIMP | 58.87623529 | **59.26882353** |
| Means | OPENLLM | 29.9997 | **30.05875** |

Table 3: Benchmark numbers for Pythia-160m pretrained at the 10M scale comparing DENSE with DYAD-IT. Instances where DYAD-IT exceeds DENSE are marked in bold, while instances where DYAD-IT falls below 0.95* DENSE are marked in Red. DYAD-IT falls below the 0.95% mark w.r.t. DENSE on only 3 zero-shot and 2 **GLUE+** tasks, falling above the mark on all GLUE+ aggregate tasks and OPENLLM. We present aggregates for brevity and defer individual values to Appendix Table 7

| Model | Forward Pass | Backward Pass | Total | Total speedup ratio |
|---|---|---|---|---|
| Dense | 101.89 | 220.16 | 332.64 | 1 |
| DYAD-IT | 89.40 | 229.86 | 310.62 | 1.071 |

Table 4: Mean time taken per minibatch by all modules of Pythia-160m training on account of forward, backward passes and in total. Times are in milliseconds. Speedup ratio is computed w.r.t. DENSE

| Model | Forward Pass | Backward Pass | Total | Total speedup ratio |
|---|---|---|---|---|
| Dense | 1.414 | 2.826 | 4.240 | 1 |
| DYAD-IT | 1.070 | 2.879 | 3.949 | 1.074 |
| DYAD-IT-8 | 0.795 | 1.843 | 2.637 | 1.607 |

Table 5: Mean time taken per minibatch by the $ff$ (feedforward) modules of Pythia-160m training on account of forward, backward passes and in total. All times are in milliseconds. Speedup ratio is computed w.r.t. DENSE

## 4 Future Work

In the future, we aim to explore i) using a heterogeneous mix of DYAD variants to approximate different ff layers ii) Replicating our experiments other minified corpora such as Minipile [13].

**Acknowledgements:** We thank Ryan McDonald and Nirmal Mukhi (ASAPP Inc.), the workshop organizers of the WANT and ESNLP workshops as well as anonymous reviewers for their helpful feedback.

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

# 5 Appendix

## 5.1 Important Additional Caveats About Formulation & Implementation

1. **Constraints on Rectangular $W$ dimensions:** Since $n_{dyad}$ denotes the number of equi-sized blocks (with W's dimensions being factorable out at $n_{in} \times n_{dyad}$ and $n_{out} \times n_{dyad}$, for a non-trivial sparse reduction, the dimensions of W woud need to be both divisible by some $n_{dyad} > 1$ — one cannot divide a $7 \times 6$ matrix into $4 \times 4$ blocks. However, we can see that for practical usage this aspect is somewhat pedantic , one can always pad up the dimensions with zeroes to different extents such that $n_{dyad}$, i.e., the desired level of sparsity is attained - e.g, in the $7 \times 6$ case, zero-padding up the number of rows by 1, our dimensions will now have a common factor 2,

2. **Additional Kernel Launches in Implementation:** The code for DYAD-IT described in the Formulation section does have some overhead in terms of additional kernel launches but for larger sized models this overhead will amortize away.

## 5.2 Hyperparameter Choices & Compute Details

For simplicity, we avoid mixed precision training (use fp32 throughout), gradient checkpointing or quantization. Since BABYLM's training setup required using earlier versions of Pytorch than would be compatible don't use FlashAttention. These techniques are in either case not intertwined directly to our method. All our OPT-125m experiments for both the STRICT and STRSMA scales were done on a NVIDIA V100. For OPT-350m experiments, we use a A10G.

We use the optimizers for dense layers from the typical DENSE setting as-is, sans any changes particular to our method. Initialization for the various SPARSE approaches too, was done in the same fashion as for DENSE. Optimizers same

## 5.3 Additional Results

### 5.3.1 Complete Benchmark Result Tables

### 5.3.2 Complete Timing Results

### 5.3.3 Complete Memory Results

In this section, we enclose complete results of evaluating each of our experiments along the aspects of memory & parameter footprint we earmarked in §3.3

### 5.3.4 Results on 100M Scale

## 5.4 Representational power of Dyad

Consider a network with two square Dyad layers i.e. $n_{in} = n_{out}$ sequentially applied one after the other to the input. Let the weight matrixes for the layers be, $W_1^d$ and $W_2^d$ and the input be $X$. The output $Y$ can be calculated as $Y = W_2^d W_1^d X$.

Consider an input dimension $i$ of $X$ and an output dimension $j$ of $Y$. If $i//n_{in} = j//n_{in}$ i.e. they fall in the same block of the BLOCKDIAG then there exists $O(n_{in})$ connections between them through the middle layer. If $i//n_{in} \neq j//n_{in}$ then only through the BLOCKDIAG there wouldn't be any interactions between this pair of input and output. However, the BLOCKTRANS interacts with outputs that are spaced uniformly apart at a stride of $n_{dyad}$. On average $O(n_{in}/n_{dyad})$ fall in the same block as that of the output dimension $j$, i.e. if the middle dimension was $k$ then $k//n_{in} = j//n_{in}$. Each of these middle dimensions will have a direct connection to j. Thus, in this second case the input dimension $i$ will have $O(n_{in}/n_{dyad})$ connections to output dimension $j$. This is summarized in Eq 17.

$$\text{No. of connections in Dyad} = \begin{cases} O(n_{in}), & \text{if } j//n_{in} = i//n_{in} \\ O(n_{in}/n_{dyad}), & \text{otherwise} \end{cases} \tag{17}$$

In the case of a sequence of two dense linear layers with the same shape, the number of connections would be $O(n_{in} \times n_{dyad})$ between each input and output. Thus, the ratio of connections between dense and linear are as shown in Eq 18.

| Benchmark | Task | DENSE | DENSE-EXT | DYAD-IT | DYAD-OT | DYAD-DT | DYAD-IT-8 |
|---|---|---|---|---|---|---|---|
| GLUE+ | CoLA | 68.50 | 64.60 | 68.20 | 68.11 | 67.32 | 67.42 |
| | SST-2 | 86.42 | 81.90 | 86.61 | 85.83 | 85.04 | 85.24 |
| | MPRC (F1) | 76.56 | 72.50 | 77.44 | 76.98 | 78.49 | 73.56 |
| | QQP (F1) | 80.50 | 60.40 | 79.79 | 80.26 | 80.91 | 80.85 |
| | MNLI | 70.77 | 57.60 | 71.12 | 71.32 | 70.89 | 70.82 |
| | MNLI-mm | 71.80 | 60.00 | 72.06 | 72.52 | 72.57 | 70.99 |
| | QNLI | 69.90 | 61.50 | 70.91 | 76.73 | 74.67 | 70.21 |
| | RTE | 60.61 | 60.00 | 48.49 | 52.53 | 56.57 | 58.59 |
| | BoolQ | 66.25 | 63.30 | 64.32 | 63.90 | 63.62 | 64.18 |
| | MultiRC | 56.30 | 55.20 | 57.07 | 57.94 | 48.96 | 54.33 |
| | WSC | 49.40 | 60.20 | 44.58 | 46.99 | 55.42 | 48.19 |
| GLUE+ Means | GLUE+ | 68.82 | 63.38 | 67.33 | 68.46 | 68.59 | 67.70 |
| | GLUE+-QA | 66.37 | 63.67 | 66.27 | 66.27 | 63.69 | 64.02 |
| | GLUE+-NLI | 68.27 | 59.78 | 65.64 | 68.27 | 68.67 | 67.65 |
| BLIMP | Anaphor Agr. | 49.49 | 63.80 | 67.33 | 64.88 | 73.93 | 59.25 |
| | Agr. Structure | 68.10 | 70.60 | 71.34 | 68.47 | 68.65 | 67.82 |
| | Binding | 68.67 | 67.10 | 65.95 | 65.18 | 63.60 | 68.94 |
| | Control/Raising | 66.64 | 66.50 | 63.52 | 64.27 | 63.83 | 62.73 |
| | D-N Agr. | 74.20 | 78.50 | 81.05 | 81.25 | 80.15 | 74.25 |
| | Ellipsis | 57.33 | 62.00 | 61.09 | 57.51 | 54.22 | 55.08 |
| | Filler-Gap | 65.36 | 63.80 | 64.58 | 65.64 | 66.67 | 65.95 |
| | Irregular Forms | 77.66 | 67.50 | 82.75 | 75.88 | 81.78 | 66.62 |
| | Island Effects | 44.02 | 48.60 | 54.75 | 49.89 | 47.35 | 48.28 |
| | NPI Licensing | 41.19 | 46.70 | 47.46 | 42.96 | 49.59 | 39.31 |
| | Quantifiers | 61.57 | 59.60 | 53.66 | 71.46 | 44.87 | 67.13 |
| | Subject-Verb Agreement | 54.62 | 56.90 | 55.77 | 61.12 | 56.95 | 63.88 |
| | Hypernym | 49.19 | 50.00 | 48.72 | 46.74 | 49.30 | 50.70 |
| | QA Congruence (Easy) | 57.81 | 54.70 | 59.38 | 60.94 | 54.69 | 57.81 |
| | QA Congruence (Tricky) | 32.73 | 31.50 | 35.758 | 47.88 | 39.39 | 39.39 |
| | Subject Auxiliary Inversion | 73.92 | 80.30 | 56.01 | 70.77 | 72.92 | 73.506 |
| | Turn Taking | 63.21 | 57.10 | 58.93 | 68.57 | 66.79 | 55.00 |
| Means | BLIMP | 59.16 | 60.31 | 60.47 | 62.55 | 60.86 | 58.88 |
| OPENLLM | ArcChallenge-25 | 22.78 | 23.72 | 22.87 | 25.26 | 23.29 | 23.293 |
| | Hellaswag-10 | 25.81 | 25.11 | 25.16 | 24.77 | 24.80 | 25.43 |
| | TruthfulQA-MC-0 | 49.39 | 49.72 | 51.12 | 48.83 | 49.84 | 49.68 |
| | MMLU-5 | 23.11 | 23.01 | 23.30 | 24.10 | 24.40 | 24.20 |
| Means | OPENLLM | 30.27 | 30.39 | 30.61 | 30.74 | 30.58 | 30.65 |

Table 6: Performance on GLUE+ (post-finetuning), BLIMP (zero-shot), OPENLLM (few-shot) benchmarks for the DENSE and DENSE-EXT baselines and all 3 Dyad variants as well as a doubly sparser version of the 1st variant. These results are with OPT-125m when pretrained at the 10M scale, a summary of which is presented in the results - the rows corresponding to Benchmark aggregate means from this table were presented in Table 1 of the main paper.

$$\text{Ratio of connections in Linear to Dyad} = \begin{cases} O(n_{dyad}), & \text{if } j//n_{in} = i//n_{in} \\ O(n_{dyad}^2), & \text{otherwise} \end{cases} \quad (18)$$

Hence, Dyad layer has the ability to mix dimensions that are both near by and far away but the ability to mix information in nearby dimensions falls linearly with sparsity but far away dimensions fall quadratically. This means that Dyad will have a bias for pushing information that needs to interact with each other a lot close by and thus more efficiently using it's parameter space when compared to linear layers. Also the inter connections between the input and output dimensions fall gradually with $n_{dyad}$ and thus provides a way to tradeoff between representational power and computational cost.

| Benchmark | Task | DENSE | DYAD-IT |
|---|---|---|---|
| GLUE+ | CoLA | 70.069 | 69.48 |
| | SST-2 | 85.039 | 85.039 |
| | MPRC (F1) | 80.435 | 79.715 |
| | QQP (F1) | 81.125 | 81.356 |
| | MNLI | 71.853 | 70.908 |
| | MNLI-mm | 73.297 | 71.929 |
| | QNLI | 80.315 | 76.859 |
| | RTE | 53.535 | 43.434 |
| | BoolQ | 64.73 | 64.315 |
| | MultiRC | 49.726 | 50.383 |
| | WSC | 53.012 | **59.036** |
| Means | GLUE+ | 69.376 | **72.220** |
| | GLUE+-QA | 64.964 | 64.804 |
| | GLUE+-NLI | 69.750 | **73.232** |
| BLIMP | Anaphor Agr. | 62.168 | 60.685 |
| | Agr. Structure | 69.241 | 67.083 |
| | Binding | 72.069 | 66.014 |
| | Control/Raising | 67.852 | 61.6 |
| | D-N Agr. | 87.019 | 84.633 |
| | Ellipsis | 62.875 | 63.279 |
| | Filler-Gap | 68.830 | 68.659 |
| | Irregular Forms | 84.173 | 73.232 |
| | Island Effects | 46.375 | **52.242** |
| | NPI Licensing | 57.060 | 46.083 |
| | Quantifiers | 68.959 | 66.718 |
| | Subject Verb Agreement | 67.66 | 59.422 |
| | Hypernym | 44.651 | **50** |
| | QA Congruence (Easy) | 54.688 | 50 |
| | QA Congruence (Tricky) | 47.879 | **50.303** |
| | Subject Auxiliary Inversion | 78.970 | 64.089 |
| | Turn Taking | 64.286 | 61.071 |
| Means | BLIMP | 64.98 | 61.47 |
| OPENLLM | ArcChallenge-25 | 23.379 | 24.500 |
| | Hellaswag-10 | 25.085 | 25.035 |
| | TruthfulQA-MC-0 | 48.661 | 50.291 |
| | MMLU-5 | 23.190 | 22.910 |
| Means | OPENLLM | 30.078 | 30.680 |

Table 7: Benchmark numbers for OPT-350m pretrained at the 10M scale comparing DENSE with DYAD-IT. Instances where DYAD-IT exceeds DENSE are marked in bold, while instances where DYAD-IT falls below 0.95* DENSE are marked in Red. We can see this happens only for four zero-shot tasks and none of the few-shot tasks.

| Benchmark | Task | Dense | Dyad-IT |
|---|---|---|---|
| | CoLA | 68.4 | **68.597** |
| GLUE+ | SST-2 | 85.236 | 84.843 |
| | MPRC (F1) | 78.873 | 78.261 |
| | QQP (F1) | 80.336 | 80.54 |
| | MNLI | 70.451 | 69.918 |
| | MNLI-mm | 70.321 | **70.974** |
| | QNLI | 55.118 | **73.447** |
| | RTE | 48.485 | 44.444 |
| | BoolQ | 66.113 | 65.422 |
| | MultiRC | 55.75 | 51.698 |
| | WSC | 42.169 | **53.012** |
| Means | GLUE+ | 73.86818182 | **73.71942857** |
| | GLUE+-QA | 69.875 | **72.7185** |
| | GLUE+-NLI | 73.188 | **77.2675** |
| BLIMP | Anaphor Agr. | 56.851 | 55.419 |
| | Agr. Structure | 68.671 | **68.708** |
| | Binding | 67.854 | 64.619 |
| | Control/Raising | 58.948 | **59.677** |
| | D-N Agr. | 75.55 | **75.961** |
| | Ellipsis | 62.356 | 60.624 |
| | Filler-Gap | 59.835 | **61.656** |
| | Irregular Forms | 56.132 | **57.30**3 |
| | Island Effects | 51.345 | **53.812** |
| | NPI Licensing | 48.558 | **55.421** |
| | Quantifiers | 60.768 | 58.733 |
| | Subject Verb Agreement | 58.229 | 53.64 |
| | Hypernym | 49.07 | **52.791** |
| | QA Congruence (Easy) | 53.125 | **57.812** |
| | QA Congruence (Tricky) | 39.394 | **51.515** |
| | Subject Auxiliary Inversion | 68.139 | 61.308 |
| | Turn Taking | 66.071 | 58.571 |
| Means | BLIMP | 58.87623529 | **59.26882353** |
| OpenLlm | ArcChallenge-25 | 23.549 | 22.696 |
| | Hellaswag-10 | 26.260 | 25.423 |
| | TruthfulQA-MC-0 | 46.972 | **48.618** |
| | MMLU-5 | 23.218 | **23.498** |
| Means | OpenLlm | 29.9997 | **30.05875** |

Table 8: Benchmark numbers for Pythia-160m pretrained at the 10M scale comparing DENSE with DYAD-IT. Instances where DYAD-IT exceeds DENSE are marked in bold, while instances where DYAD-IT falls below 0.95* DENSE are marked in Red. DYAD-IT falls below the 0.95% mark w.r.t. DENSE on only 3 zero-shot and 2 **GLUE+** tasks, falling above the mark on all GLUE+ aggregate tasks and OPENLLM

| Model | Forward Pass | Backward Pass | Total | Total speedup ratio |
|---|---|---|---|---|
| Dense | 96.57443477 | 218.1589193 | 315.6306277 | 1 |
| Dyad-IT-4 | 83.38802419 | 208.3585416 | 292.6851179 | 1.078396572 |
| Dyad-OT-4 | 82.48964827 | 207.7835725 | 291.2115524 | 1.083853388 |
| Dyad-DT-4 | 83.34073742 | 210.0591608 | 294.3693217 | 1.072226636 |
| Dyad-IT-8 | 78.16424509 | 194.1724526 | 273.3341317 | 1.154742826 |

Table 9: Mean time taken per minibatch by all modules of OPT-125m training on account of forward, backward passes and in total. All times are in milliseconds. Speedup ratio is computed w.r.t. DENSE

| Model | Forward Pass | Backward Pass | Total | Total speedup ratio |
|---|---|---|---|---|
| Dense | 2.548222502 | 4.971815463 | 7.520037964 | 1 |
| Dyad-IT-4 | 1.744403627 | 3.747922349 | 5.492325977 | 1.369190029 |
| Dyad-IT-8 | 1.111917225 | 3.026367151 | 4.138284376 | 1.817187337 |

Table 10: Mean time taken per minibatch by the ff transformer modules of OPT-350m training on account of forward, backward passes and in total. All times are in milliseconds. Speedup ratio is computed w.r.t. DENSE

| Model | Checkpoint Size (MB) | # Params | In-Train GPU Use (MB) | % Drop In GPU Mem vs Dense |
|---|---|---|---|---|
| Dense | 478 | 86.63 | 9838 | 0 |
| Dyad-IT-4 | 370 | 58.32 | 9666 | 1.74832283 |
| Dyad-OT-4 | 370 | 58.32 | 9666 | 1.74832283 |
| Dyad-DT-4 | 370 | 58.32 | 9666 | 1.74832283 |
| Dyad-IT-8 | 316 | 44.16 | 9540 | 3.029070949 |

Table 11: Mem./Param. Usage Metrics Across DENSE and other Dyad variants for OPT-125m

| Benchmark | Task | DENSE | DENSE-EXT | DYAD-IT |
|---|---|---|---|---|
| GLUE+ | CoLA | 76.742 | 73.7 | 74.877 |
| | SST-2 | 87.992 | 86.6 | 89.567 |
| | MPRC (F1) | 82.129 | 82.1 | 80.292 |
| | QQP (F1) | 83.993 | 77.8 | 82.151 |
| | MNLI | 77.339 | 70.1 | 76.623 |
| | MNLI-mm | 78.326 | 71.9 | 77.912 |
| | QNLI | 83.552 | 80.1 | 84.208 |
| | RTE | 53.535 | 67.7 | 63.366 |
| | BoolQ | 65.284 | 66.0 | 65.145 |
| | MultiRC | 62.212 | 61.1 | 64.294 |
| | WSC | 61.446 | 59.0 | 59.036 |
| Means | GLUE+ | 73.5808 | 72.24 | 74.2594 |
| | GLUE+-QA | 69.875 | 69.7333 | 69.91033 |
| | GLUE+-NLI | 73.188 | 72.45 | 75.52725 |
| BLIMP | Anaphor Agr. | 97.90 | 94.9 | 90.03 |
| | Agr. Structure | 77.885 | 73.8 | 78.48 |
| | Binding | 72.306 | 73.8 | 73.89 |
| | Control/Raising | 74.105 | 72.2 | 72.67 |
| | D-N Agr. | 93.039 | 93.1 | 91.46 |
| | Ellipsis | 81.062 | 80.5 | 81.64 |
| | Filler-Gap | 74.214 | 73.6 | 74.167 |
| | Irregular Forms | 89.924 | 80.8 | 85.55 |
| | Island Effects | 62.780 | 57.8 | 57.81 |
| | NPI Licensing | 61.160 | 51.6 | 50.42 |
| | Quantifiers | 71.303 | 74.5 | 67.46 |
| | Subject Verb Agreement | 82.240 | 77.3 | 78.64 |
| | Hypernym | 47.791 | 46.3 | 47.56 |
| | QA Congruence (Easy) | 70.312 | 76.5 | 76.56 |
| | QA Congruence (Tricky) | 52.121 | 47.9 | 50.91 |
| | Subject Auxiliary Inversion | 85.045 | 85.3 | 83.85 |
| | Turn Taking | 79.643 | 82.9 | 79.28 |
| Means | BLIMP | 74.8723 | 73.1058 | 72.9633 |
| OPENLLM | ArcChallenge-25 | 25.256 | 23.293 | 24.659 |
| | Hellaswag-10 | 25.234 | 25.055 | 25.473 |
| | TruthfulQA-MC-0 | 48.868 | 48.448 | 49.332 |
| | MMLU-5 | 23.567 | 23.181 | 23.080 |
| Means | OPENLLM | 30.73 | 29.99 | 30.636 |

Table 12: Benchmark numbers for OPT-125m pretrained on STR (100M) comparing internal and external DENSE baselines with Layer Variant

