# OpenReview forum: "DYAD: A Descriptive Yet Abjuring Density efficient approximation to linear neural network layers"
_NeurIPS.cc/2023/Workshop/WANT — WANT@NeurIPS 2023 Poster_

### Official Review · Reviewer_im4F · 2023-10-23
**I suggest weak accept, but numerical performance must be studied thoroughly**

**Confidence:** 4

**Review:**

The paper proposes yet another surrogate for the linear layer. The proposed approach relies on a hard-coded permutation matrix $P$ and can be simplified with the following equations:

1. Create a permuted copy of input $\hat{X} = PX$

2. Multiply by 4 different block-diagonal matrices:

    a. $Y_1 = W_1 X$

    b. $Y_2 = W_2 \hat{X}$

    c. $Y_3 = W_3 X$

    d. $Y_4 = W_4 \hat{X}$

3. Transpose two of 4 intermediate outputs and sum them up:

    $Y = Y_1 + Y_2 + P(Y_3+Y_4)$


The approach itself seems very easy yet even more promising. Unfortunately, the authors focus on a reduced number of floating point operations without even thinking of memory throughput. The role of memory speed in total model throughput must be investigated properly. What if size of weight matrix W of a linear layer is smaller than a size of an input activation? In certain circumstances, most of time will be spent reading input X and writing output Y, not dealing with W. And your code literally creates two tensors of the same shape as output Y. The same is true about permuted input X. Hence, read-write time doubles. And in situations, where memory is the bottleneck throughput becomes even lower. And, of course, temporary x2 and intermediate batched gemm create additional pressure on memory. Overall, such an approach can be applied ONLY in the case weight matrix W is much larger than input X and matrix multiplications are in compute-bound regime. Please provide an extensive study of applicability, with several models of different sizes and many different batch sizes. I believe for «small» models and «large» batch sizes the proposed approach will simply have a smaller throughput compared to a standard dense layer.

Minor mistakes are listed here:
1. It is hard to understand from the text. that n_in is f_in/n_dyad.  The notation makes me feel like n_in is an alias for f_in. This makes it hard reading equations, as all the reshapes seem to enlarge data by an axis broadcasting. Rename n_in and n_out such that it could be deduced from the notation that it is a size of non-zero block.
2. Table 1 is not referenced in the paper
3. Line 217: Tables ??
4. Table 3. Marked in RED… But there is no red text in the table.
5. Line 328. «We use A10G». IS it NVidia A10 with 24GB onboard? Seems like letter «G» is excess.
6. Section 5.3 is empty, why would one keep it? Especially section 5.3.3, which is referred from section 3.3.
7. Section 5.4, from my point of view, is unnecessary.
8. Table 11 did not fit into the PDF.

---

### Official Review · Reviewer_Acde · 2023-10-25
**Simple idea but suffers from quality loss in some cases.**

**Confidence:** 3

**Review:**

This paper proposes a simple idea to speed up the dense layer (MLP), which turns out to be quite heavy in modern transformer models. The proposed approach rewrites the matrix multiplication in a 3D shape and perform batched matrix multiplication for speed up. The idea is simple and easy to implement. However the following concerns are to be addressed:

1. It seems there could be some quality loss in the models trained with DYAD, which seems to be resulting from the sparsity assumption of the matrices. This seems not quite desirable in reality.
2. It would be interesting to compare the results with matrix factorization techniques that can also reduce multiplication runtime.

---

### Official Review · Reviewer_kMHp · 2023-10-27
**DYAD layer - memory-efficient alternative to Dense layer, but poorly explained**

**Confidence:** 4

**Review:**

### Summary

The paper introduces DYAD layer, alternative to Dense layer and provides experiments with them.

### Strengths

The paper offers a theoretical foundation for the structure of DYAD layers, enhancing its credibility.

### Weaknesses

The results section is disorganized; tables and figures are not located near the relevant text.

There are omissions, including a missing table at line 217 and a missing section 5.3 in the appendix.

The paper lacks clarity in certain areas, such as the unspecified metric in section 3.4.1 ("<= 5% of what?").

The paper contains grammatical errors that hinder readability.

### Recomendations
Consider replacing the PyTorch implementation of DYAD with pseudocode to focus on essential details.

Adding figures to sections 2.2.1 and 2.2.2 could clarify the processes of reshaping and block multiplication.

---

### Meta-Review · Area_Chair_MJvR · 2023-10-27

**Recommendation:** Accept (Poster)
**Confidence:** 4

**Metareview:**

**Strengths:**
* Paper targets a relevant problem and proposes a simple and easy to implement approach.

**Weaknesses:**
* No comparison to matrix factorization techniques.
* Limited applicability: approach appears to be practical only when the weight matrix is much larger than input tensor.

It is not entirely clear from the reviews if this paper should be accepted. There were some serious concerns raised about applicability, and it’s not clear from the reviews if the evaluation is strong/sufficient. I recommend rejection.

Edit: changing to accept based on additional discussions with chairs and reviewers. We think that the ideas presented in the paper are interesting and worth discussing at the workshop, although the evaluation and results need further work.

---

### Decision · Program_Chairs · 2023-10-28

**Decision:**

Accept (Poster)

**Comment:**

We thank the authors for their time and contribution to WANT and we are pleased to share that after the reviewing process the paper has been accepted. Congratulations! We encourage the authors to consider reviewers' feedback for the improvement of the camera-ready version. We hope to see you in person at the workshop and brainstorm on efficient training research together!